# Photonic-electronic integrated circuit-based coherent LiDAR engine

Anton Lukashchuk [1,4], Halil Kerim Yildirim [2,4], Andrea Bancora [1], Grigory Lihachev[1], Yang Liu [1], Zheru Qiu [1], Xinru Ji[1], Andrey Voloshin[1], Sunil A. Bhave [3], Edoardo Charbon [2] ✉ & Tobias J. Kippenberg [1] ✉

Chip-scale integration is a key enabler for the deployment of photonic technologies. Coherent laser ranging or FMCW LiDAR, a perception technology that benefits from instantaneous velocity and distance detection, eye-safe operation, long-range, and immunity to interference. However, wafer-scale integration of these systems has been challenged by stringent requirements on laser coherence, frequency agility, and the necessity for optical amplifiers. Here, we demonstrate a photonic-electronic LiDAR source composed of a micro-electronic-based high-voltage arbitrary waveform generator, a hybrid photonic circuit-based tunable Vernier laser with piezoelectric actuators, and an erbium-doped waveguide amplifier. Importantly, all systems are realized in a wafer-scale manufacturing-compatible process comprising III-V semiconductors, silicon nitride photonic integrated circuits, and 130-nm SiGe bipolar complementary metal-oxide-semiconductor (CMOS) technology. We conducted ranging experiments at a 10-meter distance with a precision level of 10 cm and a 50 kHz acquisition rate. The laser source is turnkey and linearization-free, and it can be seamlessly integrated with existing focal plane and optical phased array LiDAR approaches.

Laser ranging (LiDAR) is a widespread perception technology that is rapidly developing using recent progress in silicon photonics[1–3]. LiDAR is ubiquitous in robotics, spatial mapping, and AR/VR applications and gained popularity in the early 2000s as a key enabler of autonomous vehicles in urban environments, a goal highlighted by DARPA Grand Challenges[4]. Widely employed in the early 2000's time-of-flight sensors, which measure the arrival time of reflected pulses, relied on available legacy 900 nm diode lasers and silicon detectors. Another type of LiDAR is frequency-modulated continuous wave (FMCW) LiDAR[5,6], which maps the distance and velocity of an object to frequency. This method, an optical analogue of coherent RADAR, utilizes optical self-heterodyne detection of a frequency-chirped continuous-wave light reflected from a target with its replica, that serves as the local oscillator (LO). In contrast to the time-of-flight approach, coherent ranging allows for instantaneous velocity measurement via

the Doppler frequency shift, quantum noise limited detection enabled by heterodyne detection with sufficient LO power, eye-safe operation at low average powers, immunity to ambient light sources, and low-bandwidth receiver electronics (100s of MHz) capable of providing cm-level resolution mainly dependent on frequency excursion of the transmitted chirp. However, the cost and bulky size of individual LiDAR components and their assembly still preclude the wide adoption of ranging sensors.

Frequency-modulated continuous wave LiDAR, in particular, requires multiple building blocks, including a frequency-agile laser, driving electronics, scanning optics, passive components (grating couplers, switching network), and detectors. A variety of recent work attempted to integrate coherent LiDAR components on chip. Martin et al. demonstrated a silicon photonic circuit with integrated detectors, waveform calibration and switching network for passive beam

---

[1]Institute of Physics, Swiss Federal Institute of Technology Lausanne (EPFL), CH-1015 Lausanne, Switzerland. [2]Advanced Quantum Architecture Laboratory (AQUA), Swiss Federal Institute of Technology Lausanne (EPFL), CH-2002 Neuchâtel, Switzerland. [3]OxideMEMS Lab, Purdue University, 47907 West Lafayette, IN, USA. [4]These authors contributed equally: Anton Lukashchuk, Halil Kerim Yildirim. ✉e-mail: edoardo.charbon@epfl.ch; tobias.kippenberg@epfl.ch

scanning capable of 60 m coherent ranging at 5 mW output power[7]. A number of recent works employed optical phased array (OPA) technology to achieve 2D passive scanning[8–10]. Poulton et al. demonstrated a nearly centimeter scale OPA aperture with 8192 elements achieving $100^o \times 17^o$ field of view[3]. Rogers et al. developed a focal plane array (FPA) 3D LiDAR on a silicon chip with photonic-electronic monolithic integration of a 512-pixel coherent receiver array[1]. Further large scaling of FPA pixels employing switchable MEMS grating antennas was reported by Zhang et al.[11].

The aforementioned approaches are CMOS compatible, can be integrated with other passive or active optical components and are scalable, i.e., support a further increase in the number of pixels and field of view. However, these prior demonstrations all used external lasers coupled via fiber, off-the-shelf driving electronics, and bulk fiber-based erbium-doped amplifiers for signal or reflection amplification and bulk modulators (Refs. 1,8,12) - significantly compounding full integration. A fully integrated FMCW LiDAR will require to address the remaining integration and replace these building blocks with their photonic integrated circuit-based counterparts. Tackling the issue of discrete external components, Isaac et al. fabricated an integrated transceiver module on the InP platform[13], but no coherent ranging functionality was performed. Ref. 14 showed a fully integrated coherent LiDAR on the chip, though it is limited to single-pixel imaging only. Here, we make a step further towards fully photonic-electronic integrated LiDAR presenting an integrated laser, high-voltage arbitrary waveform generator (HV-AWG) application-specific integrated circuit (ASIC), and chip-scale Erbium-doped waveguide amplifier (EDWA) - key components for integrated LiDAR source - which are all fabricated with foundry compatible wafer scale manufacturing. Using these photonic integrated circuit-based components we demonstrated ranging at 10 m with cm-level precision. The combination of integrated laser, HV-AWG ASIC and chip-scale Erbium amplifier constitutes a robust, coherent LiDAR source, which can be applied to existing silicon imaging 3D sensors[1,2,12] and pave a path towards a fully integrated coherent LiDAR system.

## Results
### Photonic-electronic LiDAR source
The photonic-electronic LiDAR consists (cf. Fig. 1a) of three main building blocks: laser, ASIC, and on-chip amplifier. Generally, a distributed feedback laser (DFB) is used as the light source in FMCW LiDAR implementations[15–17]. While DFB lasers offer tunability with MHz actuation bandwidth and excursions up to 100s of GHz, they suffer from the need for continuous feedback for chirp linearity[18] – the linearity-phase noise and wide tunability trade-off inherent to conventional lasers[19]. Recent advances in integrated photonics have enabled to achieve fiber laser coherence, by using self-injection locking of a DFB laser to ultra-low propagation loss photonic integrated circuits based on silicon nitride[20]. By endowing such circuits with piezoelectrical MEMS actuators, it has been possible to achieve both high coherence as well as fast tuning with low nonlinearity[21], allowing linearization-free FMCW LiDAR – yet requiring large voltage driving. Other approaches based on electro-optic integrated photonic laser feedback circuits using $\chi^{(2)}$ materials, such as lithium niobate (LiNbO₃) or barium titanate (BTO)[22–24] can lead to even faster frequency chirps at a lower voltage, but presently exhibit far lower laser coherence. We note Zhang et al.[25] have recently demonstrated that such materials exhibit significantly lower levels of cavity noise at high offset frequencies and are predominantly limited by thermal-charge-carrier-refractive noise.

In our work, we employ a Vernier ring filter-based external cavity hybrid integrated laser[26]. Such laser configuration has advanced in recent years demonstrating low linewidth[26,27], fast tuning[22] and implementations on different platforms[22]. We acknowledge the existence of alternative approaches utilizing ring self-injection locking,

Bragg gratings, (extended) distributed Bragg reflectors, and vertical cavity surface emitting lasers. Performance comparison of these lasers is summarized elsewhere (Supplementary material[21,28]).

The laser (cf. Fig. 1c) operates at 1566 nm and includes a reflective semiconductor optical amplifier (RSOA) edge-coupled to a Si₃N₄ photonic integrated circuit with a microresonator-based Vernier filter. This approach has the distinct advantage of using cost-effective III-V based RSOA, that does not require gratings as in the case of DFB. Two microresonators with loaded cavity linewidths of 200 MHz slightly differ in free spectral range (FSR), i.e., 96.7 GHz and 97.9 GHz with Vernier frequency of 8.7 THz. Both rings have integrated microheaters realized in the bottom Pt electrode layer. We aligned the pair of resonances using a microheater and obtained up to 20% reflection back to the RSOA, using the feedback circuit depicted in Fig. 1a. Microheaters provide at least 100 GHz continuous cavity frequency tuning, which results in $FSR^2/\Delta FSR \approx 60$ nm discrete laser frequency tuning by matching the resonances of the Vernier ring filter and adjusting the intracavity phase[29]. Piezoelectric lead zirconate titanate (PZT) actuators were heterogeneously integrated on top of the microrings to perform fast actuation via stress-optic effect for rapid laser frequency tuning. We note that PZT can be monolithically integrated on silicon in a standard CMOS foundry process[30]. Platinum electrodes (cf. Fig. 1d) match the resonator radius to maximize the stress-optic tuning efficiency, attaining ~130 MHz/V of frequency tuning. We note that decreasing the radius of the microring with the PZT actuator improves the tuning efficiency up to 500 MHz/V as reported in[21]. When coupling the hybrid MEMS-based circuit to an RSOA, we observe lasing with an output power of 3 mW, side mode suppression ratio of 50 dB, and featuring frequency noise of $10^4$ Hz²/Hz at 10 kHz offset frequency, reaching the white noise floor of 127 Hz²/Hz at 6 MHz offset[29]. For better stability, the entire assembly is packaged within a butterfly 14-pin package and placed on a Peltier element, and light is coupled to a SMF output fiber. The RSOA, Peltier element, thermistor, all microheaters, and PZT actuators are all connected to the butterfly pins using electrical wirebonding. The hybrid packaging allows for turnkey laser operation, reduces laser frequency noise at offsets below 1 kHz and maintains the tuning performance after waveform predistortion and linearization.

Operating the laser in FMCW mode (cf. Fig. 1b) requires frequency tuning over several GHz, which for PZT integrated actuators necessitates voltages above 10 V, not achievable with conventional CMOS electronics. Moreover, conventional linear modulation laser tuning necessitates feedback to keep the waveform linear. It originates from the non-linear current tuning and electro-optical transduction. The optoelectronic feedback loop is usually utilized to lock the optical chirp detected via delay interferometer to an electronic reference[18,31]. Other techniques involve preliminary iterative predistortion and linearization of the waveform[17,21] or resampling of the recorded signal, all of which rely on auxiliary interferometers[32,33]. To eliminate the necessity for feedback or post-processing and overcome the voltage budget limitation, we designed and fabricated, a HV-AWG integrated circuit (see Fig. 1e) that generates a 20 V sawtooth waveform that drives the PZT actuators while being supplied with only 3.3 V. The electrical waveform is then transduced to the optical domain, resulting in a >2 GHz optical chirp excursion.

FMCW LiDAR can be implemented in various ways. Typically, it uses a triangularly chirped waveform[34], but can also use random phase code modulation[35]. Figure 1b shows the chirp waveform employed in our experiments. Linearly frequency-chirped laser light is split into a local oscillator path and signal path, with the signal path sent to the target. The reflected light is then mixed with the local oscillator and measured on a balanced photodiode. The detected range is proportional to the beatnote frequency inferred from the short-time Fourier transform of the recorded heterodyne signal. It is inversely

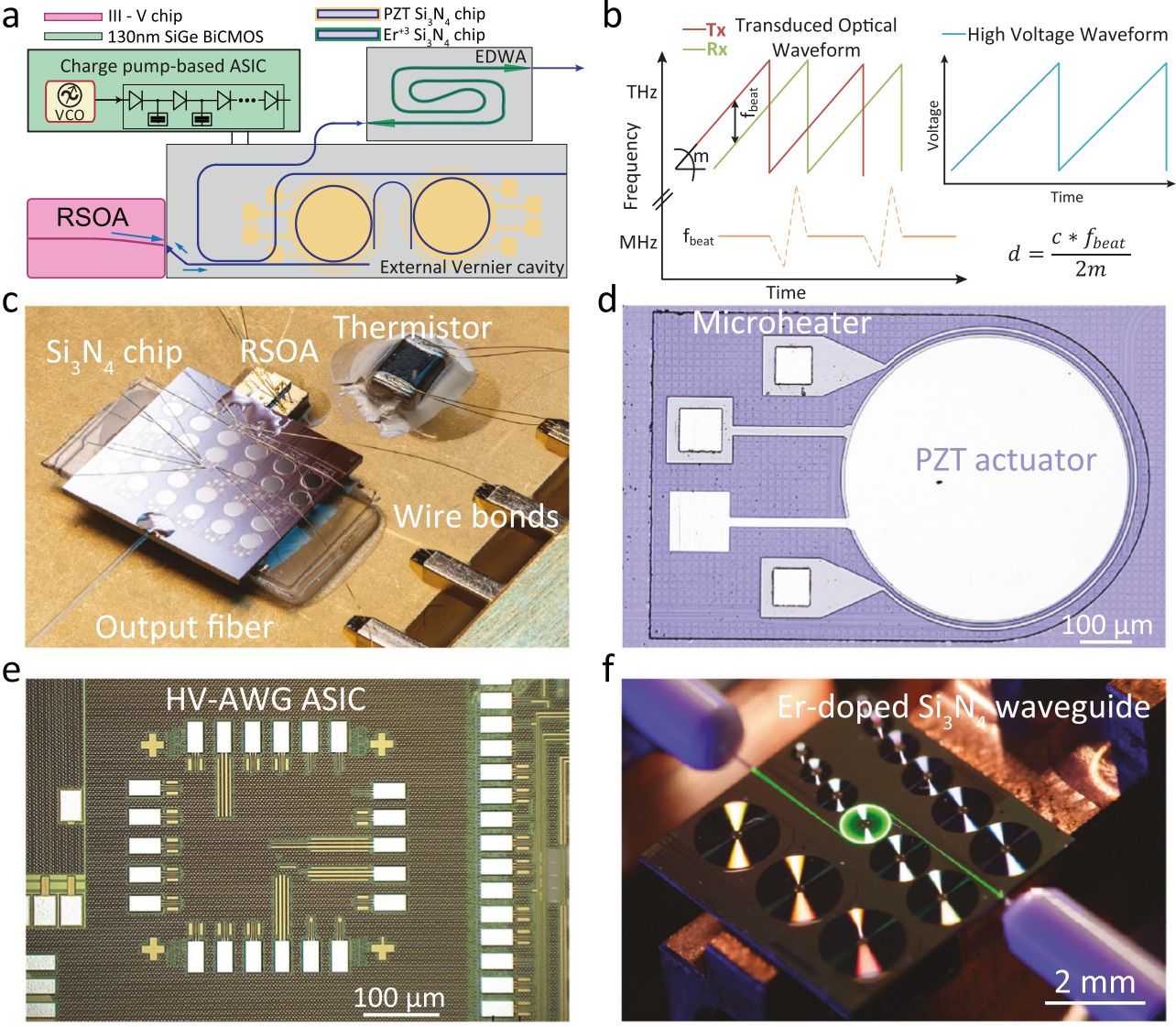

**Fig. 1 | Concept of photonic-electronic LiDAR source. a** Schematics of photonic-electronic LiDAR structure comprising a hybrid integrated laser source, charge-pump based HV-AWG ASIC, photonic integrated erbium-doped waveguide amplifier. **b** Coherent ranging principle. **c** Packaged laser source. RSOA is edge coupled to Si$_3$N$_4$ Vernier filter configuration waveguide, whereas the output is glued to the fiber port. PZT and microheater actuators are wirebonded as well as butterfly package thermistor. **d** Zoom-in view of (**c**) highlighting a microring with actuators. **e** Micrograph of the HV-AWG ASIC chip fabricated in a 130 nm SiGe BiCMOS technology. The total size of the chip is 1.17−1.07 mm². **f** The Erbium-doped waveguide is optically excited by a 1480 nm pump showing green luminescence due to the transition from a higher lying energy level to the ground state.

proportional to the waveform chirp rate $m$ - the ratio of the frequency excursion and sweeping period.

Finally, we employed a chip-scale integrated EDWA (cf. Fig. 1f) to amplify the laser to >20 mW optical power to meet the power requirements for robust and long-range coherent ranging[36]. Typically integrated LiDAR systems require > 100 mW of optical power to compensate for coupling losses and be able to emit 10s mW at the aperture[3,7]. The EDWA was implemented using an on-chip 21-cm-long Si$_3$N$_4$ spiral waveguide doped with high-concentration Erbium ions ($3.25 \times 10^{20}$ ions/cm) through a high-energy (up to 2 MeV) ion implantation process[37,38]. The doped Erbium ions can be optically pumped to the excited state and allow for amplification stemming from the stimulated transition to the ground state. The EDWA provides linear and low noise optical amplification to the frequency-modulated optical waveform due to the slow gain dynamics (millisecond lifetime) and the small emission cross section of Erbium ions[39].

## High-voltage Arbitrary Waveform Generator ASIC

Arbitrary waveform generation at high voltages is desired to drive various devices, including ultrasound transducers[40,41], piezoactuators[21], neurostimulators[42], single-photon avalanche diodes (SPADs)[43]. HV-AWGs are usually provided as single or even multiple discrete components[21,44], which are generally challenging to integrate due to their incompatibility with technologies supporting advanced electronics. We demonstrate a novel architecture, which can generate high-voltage arbitrary waveforms using a standard CMOS technology supplied at 3.3 V. Figure 2a shows the schematic block diagram of the IC. The ASIC consists of a voltage-controlled ring oscillator (VCRO), which drives the clocks of a series of Dickson charge pump stages. The oscillation frequency can be controlled externally to modify the waveform. The 11-stage charge pump generates the output waveform rising edges, whereas the 15-stage charge pump generates falling edges. The two charge pump blocks operate in a complementary fashion, with the 'Select' signal

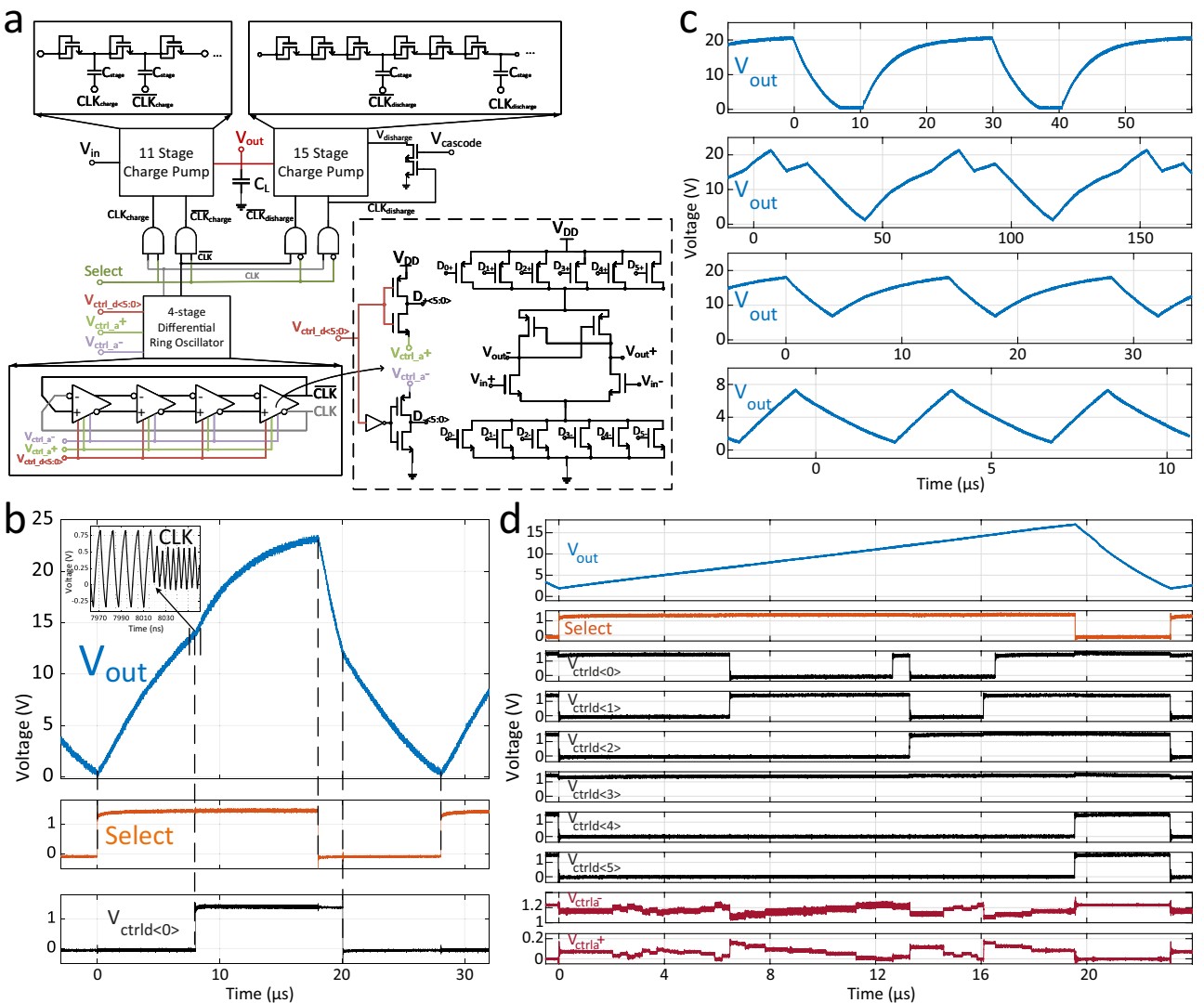

**Fig. 2 | High-voltage arbitrary waveform generator integrated circuit, fabricated in a 130-nm SiGe BiCMOS technology. a** Schematics of the integrated circuit consisting of a 4-stage voltage-controlled differential ring oscillator which drives charge pump stages to generate high-voltage arbitrary waveforms. **b** Principles of waveform generation demonstrated by the output response to the applied control signals in the time domain. Inset shows the change in oscillation frequency in response to a frequency control input, from 88 MHz to 208 MHz, which modifies the output waveform. **c** Measured arbitrary waveforms generated by the ASIC with different shapes, amplitudes, periods and offset values. **d** Generation of the linearized sawtooth electrical waveform used in LiDAR measurements. Digital and analog control signals are modulated in the time domain to fine-tune the output.

controlling whether the output voltage increases or decreases. The clocks are applied only to the charging/discharging block during a one-half cycle when the output voltage rises/falls. The oscillator is designed to have a wide frequency range and a high frequency resolution, so as to achieve fine-tuning capability while controlling the waveform. The unit cell of the four-stage ring oscillator has six geometrically sized pairs of NMOS and PMOS (n-type/p-type metal-oxide-semiconductor) transistor loads to control the unit delay. The inputs, $V_{ctrld<5:0>}$, can be digitally switched to turn on a pair of NMOS and PMOS loads each, where $V_{ctrld<5>}$ corresponds to the load with the highest width per length. Using digital inputs, the gate voltage of the PMOS loads is connected to $V_{ctrla+}$ and the NMOS loads to $V_{ctrla-}$, respectively. These two gate voltages are controlled differentially in an analog fashion, where their sum equals $V_{DD}$ to fine-tune the oscillation frequency. Operating at a supply of 1.2 V, the frequency of the designed oscillator can be set in the range of 6 MHz to 350 MHz, with a tuning control of 2% at lower frequencies and 0.5% at higher frequencies employing 10 mV steps for $V_{ctrla+}$ and $V_{ctrla-}$.

Figure 2 b illustrates the principles of waveform generation. The output voltage of a charge pump in the time domain shows an exponential response on a capacitive load, where the rise-time depends on the frequency of the applied clocks[45]. Therefore, one can tune the time-domain waveform by changing the clock frequency at pre-determined time points. We can set a digital control input of the VCRO high to decrease the waveform rise time by increasing the clock oscillation frequency, from approximately 88 MHz to 208 MHz. To start the falling edge, we switch 'Select' signal low, and the fall time is controlled in the same manner using the oscillator inputs. This allows the HV-AWG ASIC to generate output waveforms with a peak voltage of more than 20 V. The output waveform has a period similar to 'Select'; 'Select' duty cycle also sets the highest and lowest output voltage values. We can produce waveforms with different shapes, amplitudes, frequencies, and offset values when operating the circuit with different VCRO input sequences, as shown in Fig. 2c. Figure 2d shows the generated 45 kHz sawtooth waveform used in our FMCW LiDAR experiments. We linearized the exponential response of the charge pump to obtain a

sawtooth waveform. The voltage control inputs are modified in time to gradually decrease the rise-time, by changing the VCRO frequency within the range of 30-80 MHz. The digital voltages, $V_{ctrld<5:0>}$, allow a coarse control of the VCRO causing too abrupt changes in the waveform. The analog voltages, $V_{ctrla+}$ and $V_{ctrla-}$, are used in conjunction at smaller time steps, which allows fine-tuning of the optical waveform for higher linearity.

**Electro-optic transduction and linearity**

Figure 3a demonstrates the electrical waveform generated by the ASIC. The same sawtooth signal was applied to both piezoactuators of the Vernier laser. The heterodyne measurement carried out with an auxiliary laser shows the time-frequency map of the laser chirp (cf. Fig. 3b). The 15 V electrical signal resulted in >2 GHz optical frequency excursion over a 23 $\mu$s period. The chirp $m$ parameter of the up-swing used for the ranging equates to ~110 THz/s. It ultimately determines the detected beatnote frequency $f_{beat}$ to distance $d$ mapping via $d = c/2m \times f_{beat}$ where $c$ is the speed of light. In our experiment 1 m of range maps to ~1 MHz frequency beatnote for the laser sweep parameters described above.

For the long-range and robust measurement, FMCW LiDAR requires high chirp linearity of the optical waveform[17]. To obtain the required AWG signal, we iteratively linearized the optical waveform employing a delayed homodyne detection method[46]. We calculated

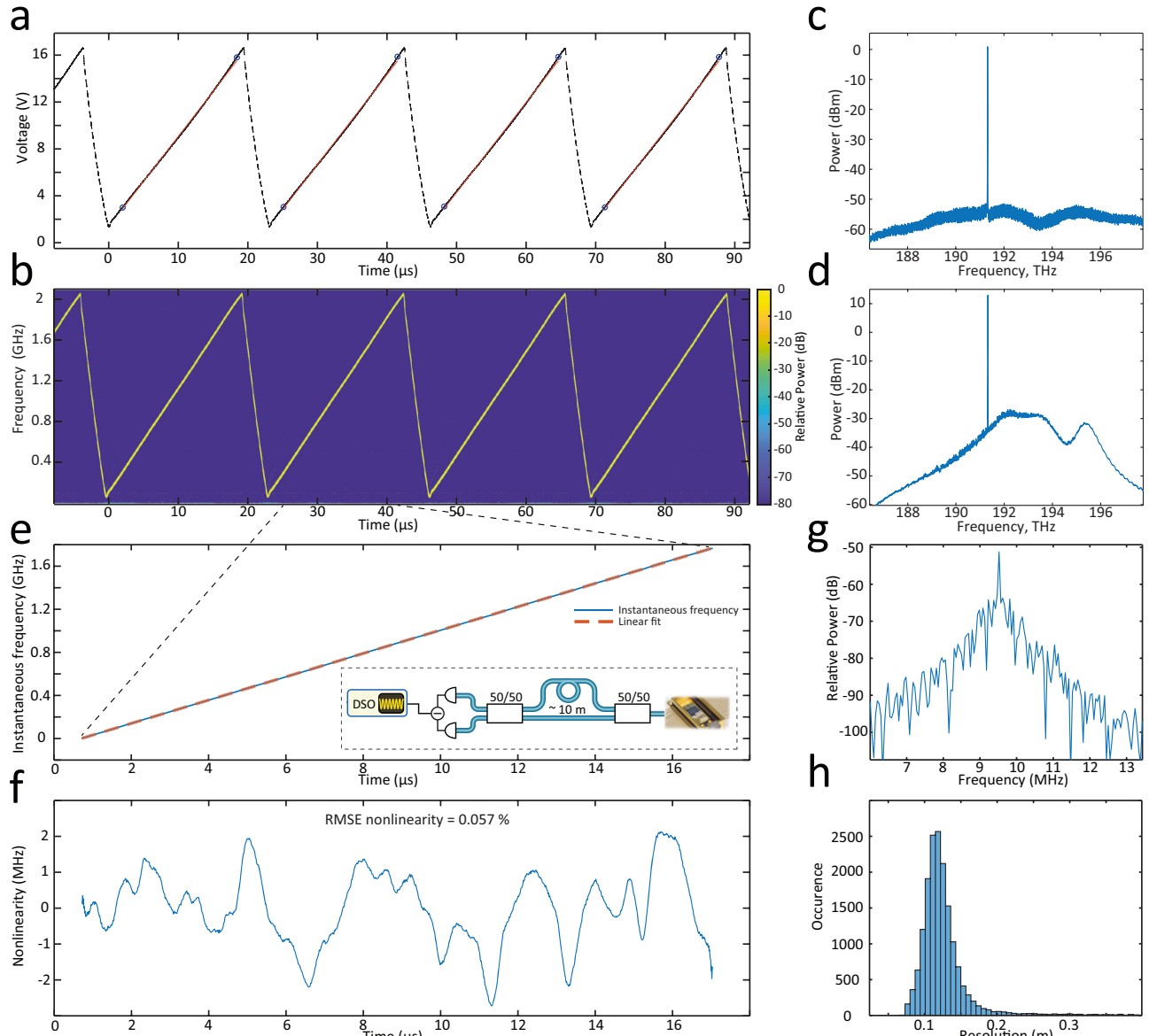

**Fig. 3 | Photonic integrated LiDAR source electro-optical transduction and linearity. a** Electrical waveform generated by the ASIC. Blue circles highlight the segment of ~16 $\mu$s used for ranging and linearity analysis. The red curve is a linear fit to the given segment. **b** Time-frequency map of the laser chirp obtained via heterodyne detection with auxiliary laser. RBW is set to 10 MHz. **c** Optical spectrum of Vernier laser output featuring 50 dB side mode suppression ratio. **d** Optical spectrum after EDWA with >20 mW optical power. **e** Instantaneous frequency of the optical chirp obtained via delayed homodyne measurement (inset: experimental setup). The red dashed line corresponds to the linear fit. The excursion of the chirp equates to 1.78 GHz over a 16 $\mu$s period. **f** Nonlinearity of the laser chirp inferred from (**e**). RMSE nonlinearity equates to 0.057% with the major chirp deviation from the linear fit lying in the window ± 2 MHz. **g** The frequency beatnote in the delayed homodyne measurement corresponds to the reference MZI delay ~10 m. The 90% fraction of the beatnote signal is taken for the Fourier transformation. **h** LiDAR resolution inferred from the FWHM of the MZI beatnotes over >20,000 realizations. The most probable resolution value is 11.5 cm, while the native resolution is 9.3 cm corresponding to 1.61 GHz (90% of 1.78 GHz).

the instantaneous frequency of the chirp (cf. Fig. 3e) via Hilbert transformation of the beatnote electric signal. The ASIC architecture allows for fine-tuning the waveform in an arbitrary fashion, therefore we optimized the frequency control inputs of the ASIC to minimize the root mean square error RMSE of the instantaneous frequency. We used a 16 $\mu$s up-rise segment of the chirp for linearity analysis. We note that we described the procedure of finding the needed waveform, which does not require any further feedback during the laser operation. This is in contrast to laser current tuning, where the feedback is essential for linear operation[18].

Figure 3f depicts the chirp nonlinearity or instantaneous frequency deviation from the fitted linear line. The major part of the deviation lies within ± 2 MHz window and exhibits a total nonlinearity of <0.1%. While we optimized the optical waveform, the voltage ramp appeared to have 0.35% linearity and 0.05 V RMSE at an overall 15 V voltage excursion due to the non-ideal electro-optic transduction. We assume the electrical waveform noise limits the linearity of the optical chirp. The VCRO has ~1000 oscillations per waveform period, with one one-step output voltage increase occurring at each cycle with charge pumping. The steps change in the range of 20 mV at low output voltages down to 8 mV at higher output voltages. This imposes a limit on

the linearity of 0.05% for 15 V chirps due to quantization error. We envision the increase in clock rate or decrease in voltage step can further improve the nonlinearity affecting ranging resolution and accuracy[17,47].

The Fourier transform of the delayed homodyne detection is shown in Fig. 3g. The full width at half maximum (FWHM) of the beatnote determines the resolution of the LiDAR. The beatnote linewidth is nearly Fourier transform limited featuring 60 kHz width. Fig. 3h presents statistics over $2 \times 10^4$ measurements. It indicates 11.5 cm resolution $\Delta R$ (most probable value) while the native resolution for $B = 1.6$ GHz (90% fraction of 1.78 GHz) excursion chirp equates to 9.3 cm following $\Delta R = c/2B$.

### Optical ranging

Figure 4a illustrates the FMCW photonic-electronic LiDAR experimental setup. The Vernier laser was turnkey initiated, and the ASIC preconfigured waveform was subsequently applied to the laser piezo-actuators. The output light was first split into the signal and local oscillator paths. The EDWA chip provided a 13 dB gain and amplified the signal up to 22 mW. The optical spectra before and after the amplification stage are depicted in Fig. 3c, d, respectively. Further

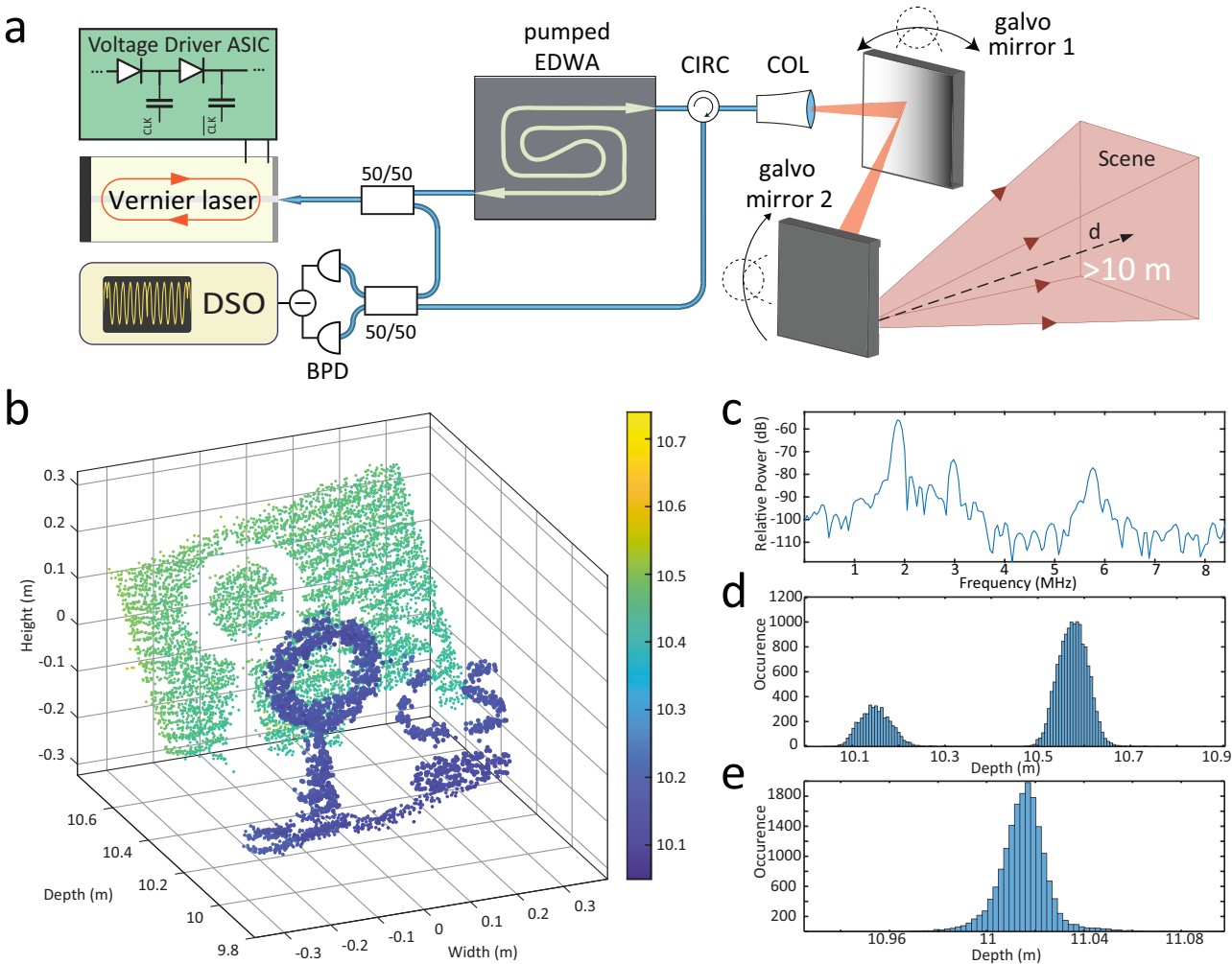

**Fig. 4 | Ranging experiment. a** Schematics of the experimental setup for ranging experiments. The amplified laser chirp scans the target scene via a set of galvo mirrors. A digital sampling oscilloscope (DSO) records the balanced detected beating of the reflected and reference optical signals. CIRC - circulator, COL - collimator, BPD - balanced photodetector. **b** Point cloud consisting of ~$10^4$ pixels featuring the doughnut on a cone and C, S letters as a target 10 m away from the collimator. **c** The Fourier transform over one period, highlighting collimator, circulator and target reflection beatnotes. Blackman-Harris window function was applied to the time trace prior to the Fourier transformation. **d** Detection histogram of (**b**). **e** Single point imaging depth histogram indicating 1.5 cm precision of the LiDAR source.

amplification is possible with double side pumping of the EDWA chip or by matching the Vernier lasing frequency to the maximum of the gain profile (Liu et al. demonstrated 24 dB off-chip amplification[38]). We employed a mono-static imaging setup with the same collimator operating as a transmitter and a receiver. Two galvo mirrors scanned the laser beam at 2 Hz vertical and 63 Hz horizontal rates. An optical circulator separated the back-reflected light from the transmitted one, whereafter it was self-heterodyne mixed on a balanced photodetector. The digital oscilloscope sampled the photocurrent at 100 MS/s during 0.25 s acquisition time.

Figure 4b shows the resulting point cloud consisting of ~10, 000 pixels. The target comprised a styrofoam doughnut and cone, C and S paper letters, and a flat background placed 10 m away from the collimator (cf. photo of the target in SI). Every pixel was obtained by analyzing a single sawtooth period of 23 μs. First, the signal was time-gated to the up-swing fraction of the chirp, and then Fourier transformed. Figure 4c depicts a periodogram of the detected signal with the Blackman-Harris window applied. The chirp excursion was obtained from the self-homodyne measurement with an auxiliary interferometer. We performed the Gaussian fitting of the target beatnote to infer the range estimate. We estimated the precision of our setup to be ~1.5 cm derived from the statistics of a single point measurement over ~$2 \times 10^4$ realizations. We note that in practical LiDAR applications, it is essential to consider real-world factors such as atmospheric turbulence, including variations in the refractive index along the beam path, as well as the influence of transverse wind[34,48].

## Discussion

In summary, we have demonstrated an integrated circuit-based coherent LiDAR source. The system-level architecture represents a drop-in frequency-agile laser with ASIC-defined FMCW laser tuning. We showed that a combination of a hybrid integrated Vernier ring laser with fast PZT actuators, a HV-AWG ASIC and an EDWA $Si_3N_4$ chip, attains 2 GHz frequency sweeps at 50 kHz rate with an output power of more than 20 mW. Our LiDAR engine achieves a 12 cm depth ranging resolution with chirp nonlinearity of less than 0.1%. Employing conventional 2D mechanical galvo mirror scanning, we demonstrated ranging at a 10 m distance with 1.5 cm precision.

The HV-AWG ASIC demonstrated in this work allows for high-voltage arbitrary waveform generation without using a high-voltage supply or an RF power amplifier. Its architecture eliminates the need of high-voltage technologies and additional discrete components, which is limiting for other state-of-the-art solutions. The charge pump-based design can generate output waveforms beyond the voltage rating of the transistors in the technology, so the driver can be implemented in advanced standard CMOS nodes. This is advantageous in terms of power consumption and computational processing capabilities of the system, but the manufacturing cost is typically higher. However, since the design has a small footprint with the possibility of having processing and a high-voltage driver on a single CMOS chip, it supports further integration with the PIC assembly, allowing further miniaturization of the LiDAR engine. Figure 1 in the SI shows the comparison of the proposed HV-AWG with other techniques in the literature, where integration, compactness and versatility are some of the advantages of the architecture. The chirp rate can be further increased from 50 kHz by reducing the parasitic capacitance at the ASIC output by minimizing the packaging or co-integration with the PIC.

Optical power amplification is commonly used in LiDAR demonstrations to achieve the required power levels, since the power output of a standalone laser is often insufficient. On-chip low-noise, high-power optical amplification has been recently made possible in Erbium-implanted $Si_3N_4$ photonic integrated circuits, showing competitive performance to widely deployed electrically pumped semiconductor optical amplifiers. In contrast to SOAs with picosecond

scale carrier lifetime, the Erbium-based amplifiers enable optical amplification of modulated optical signals with negligible gain nonlinearity or channel cross talk, and lower spontaneous emission noise, benefiting from the millisecond scale long excited state lifetime and much smaller emission cross-section, which may be crucial in presence of residual amplitude modulation of optical signals. Although we have demonstrated separate integrated components, we note that the Vernier laser and the EDWA are based on the same $Si_3N_4$ material. The laser source and the amplifier can be simultaneously integrated on the same photonic chip due to the possibility of selective area Er implantation, providing improved signal-to-noise ratio, higher output power and low fabrication complexity. We used a 1480 nm pump source for Er-ion excitation off-chip. However, it could be hybrid integrated via edge-coupling[49] or photonic wire bonded[50].

Finally, we have presented a photonic-electronic integrated coherent ranging source. Comprising integrated laser, HV-AWG ASIC and on-chip amplifier, it could be readily applied to existing LiDAR approaches[1,2,12] replacing bulk components.

## Methods

### Vernier ring filter laser operation

In our experiments, the laser operated at a temperature of 26 C$^o$ (thermistor resistance 9700 OHm) and RSOA current of 340 mA, resulting in 3 mW output optical power. We applied around 30 mW of electrical power to microheaters to align resonances of $Si_3N_4$ microrings. We do not have an on-chip phase shifter section in our laser design, and the tuning range was empirically limited to 3 GHz while the RSOA current was fixed. RSOA is edge coupled to the $Si_3N_4$ chip, where the coupling waveguide angle of the $Si_3N_4$ chip is optimized (in simulations) to achieve 20% power transmission (laser output power / nominal RSOA power).

We note that the initial linearization routine is essential for determining the electrical waveform itself and is distinct from the linearization routines applied in postprocessing or feedback. Subsequently, this waveform is stored in memory and reapplied each time. We have observed no degradation in nonlinearity during continuous laser operation over the course of a day, including when the laser is switched off and on with the same operational parameters (such as RSOA current and the current applied to the microheaters used to align the Vernier filter resonances). However, we have observed that making slight adjustments to the laser's operating point can result in a minor degradation of linearity. This effect can be mitigated, and the linearity improved to less than 0.1%, by repeating the calibration procedure.

### EDWA chip operation and fabrication

We used lensed fibres to couple the light into the EDWA chip. The 1480 nm pump laser diode was connected via an off-chip 1480/1550 nm coupler. Integrated $Si_3N_4$ spiral waveguides were fabricated with the photonic damascene process[51]. The preform for waveguide structures and filler patterns for stress release was then defined by deep ultra-violet photolithography and transferred into the thermal oxide layer using reactive ion etching (RIE). Thermal treatment at 1250° C was then applied to reflow the silicon oxide, reducing the roughness caused by the RIE etching[52], before the preform recesses were filled with stoichiometric $Si_3N_4$ by LPCVD. An etchback process was performed to roughly planarize the wafer surface and remove the excess $Si_3N_4$ material. Chemical mechanical polishing (CMP) was then applied to reach the desired waveguide thickness, and create a top surface with sub-nanometer root-mean-square roughness. Using this process, 700 nm-thick $Si_3N_4$ waveguides buried in a wet oxide cladding but without top cladding were created, which allows for direct Erbium implantation into the waveguide core. The Erbium ion beam energy of 0.955 MeV, 1.416 MeV and 2 MeV and the corresponding fluence of $2.34 \times 10^{15}$, $3.17 \times 10^{15}$ and $4.5 \times 10^{15}$ ions/cm$^2$, respectively, were consecutively applied to the separated passive $Si_3N_4$ photonic chips. This

process can implant Erbium ions into the $Si_3N_4$ waveguides $(0.7 \times 2.1 \mu m^2$ cross-section) with a maximum doping depth of ca. 400 nm from the top surface and achieve an overlap factor of ca. 50%. The doped $Si_3N_4$ was then annealed at $1000°$ C in $O_2$ under atmosphere pressure for 1 hour to heal the implantation defects and optically activate the doped Erbium ions. The measured lifetime of the first excited state of doped Erbium ions is ca. 3.4 ms. Higher annealing temperature could lead to Erbium ion precipitations in the silica cladding[37]. The Erbium ion implantation of the $Si_3N_4$ photonic chips was performed at the University of Surrey Ion Beam Centre via commercial service.

## HV-AWG ASIC operation

The design was fabricated in a standard 130-nm SiGe BiCMOS technology, where only CMOS transistors were used. The chip measures $1.17·1.07$ mm$^2$. The total active area of the design, i.e. excluding decoupling caps and IO pads, is approximately $35,000 \mu m^2$. The charge pump is implemented with diode-connected isolated thick oxide NMOS transistors and metal-insulator-metal (MiM) capacitors. $V_{in}$ and $V_{cascode}$ were biased at $V_{DD}= 3.3V$. The 15-stage charge pump has three diode-connected transistors in each stage, so as to set $V_{discharge}$ to $V_{out}$ - $45V_{th,n}$ as $V_{out}$ rises during the charging phase. A cascode transistor is also introduced at $V_{discharge}$, so that a voltage up to $2V_{DD}$ can be tolerated at this node. This ensures that $V_{out}$ can safely reach more than 20 volts without reaching the breakdown of the discharge transistors, given that the capacitors and NMOS-to-bulk isolation are within breakdown limits as well. For the measurements presented, the output of the CMOS chip was fed to an external unity-gain voltage buffer whose output was measured and/or used to drive the actuators. Load capacitance at chip output was measured at 26 pF. The printed circuit board used for controlling the ASIC is shown in SI Fig. 2.

## Data availability

Data used to produce the plots within this paper is available at https://doi.org/10.5281/zenodo.10668672. All other data used in this study are available from the corresponding authors upon request.

## Code availability

Code used to produce the plots within this paper is available at https://doi.org/10.5281/zenodo.10668672.

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

## Acknowledgements

This publication was supported by Contract W911NF2120248 (NINJA LASER) from the Defense Advanced Research Projects Agency (DARPA), Microsystems Technology Office (MTO), as well as the Swiss National Science Foundation (SNSF) through grant number 211728 (BRIDGE). A.L. acknowledges support from the European Space Technology Centre with ESA Contract No. 4000133568/20/NL/MH/hm. This work also received funding from the EU H2020 research and innovation programme under the Marie Sklodowska-Curie grant agreement No. 101033663 (RaMSoM), as well as the Horizon Europe EIC Transition grant agreement no. 101113302 (MAGNIFY).

## Author contributions

H.K.Y. designed the ASIC. A.B. and A.V. performed hybrid packaging of the laser. Y.L., Z.Q. and X.J. developed the EDWA chip. A.L. and H.K.Y. performed the measurements with assistance from A.B., G.L. and Y.L. A.L. performed the data analysis. A.L. and H.K.Y. wrote the manuscript with input from all authors. T.J.K., E.C. and S.B. supervised the work.

## Competing interests

The views, opinions and/or findings expressed are those of the authors and should not be interpreted as representing the official views or policies of the Department of Defense or the U.S. Government. Dr. Sunil Bhave performed this research at Purdue University prior to becoming a DARPA program manager. T.J.K. is a co-founder and shareholder of LiGenTec SA, a foundry commercializing Si$_3$N$_4$ photonic integrated circuits. T.J.K., A.V. are co-founders and shareholders of DEEPLIGHT SA and A.B. is a shareholder of DEEPLIGHT SA, a start-up company commercializing Si$_3$N$_4$ photonic integrated circuits based frequency agile low noise lasers. There are no other competing interests.
