## [Peer Review File · Nature Communications]

Photonic-electronic integrated circuit-based coherent LiDAR engineREVIEWER COMMENTS

Reviewer #1 (Remarks to the Author):

Lukashchuk et al. report an integrated system of frequency-chirped laser for FMCW. Narrow linewidth lasers with fast and large excursion frequency chirping are necessary for coherent FMCW LIDAR but remain to be a challenge. The authors attack this challenge with an impressive engineering endeavor that integrates multiple components, including a semiconductor optical amplifier (SOA), SiN Vernier cavities with PZT piezoelectric MEMS tuning, an erbium-doped waveguide amplifier (EDWA), and a SiGe BiCMOS. The achieved performance is equally impressive, with a frequency excursion of 2 GHz in 23 microseconds, leading to a ranging resolution of 1.5 cm. These performances are on par with the state-of-the-art FMCW demonstration realized with discrete optical and electronic components and systems. The demonstrated results are compelling and self-evident and the analysis is carefully done with standard approaches, so there are really no technical comments I can provide. Although, in my opinion, the claim that the system is "plug-and-play" is an exaggeration, as the whole system is very complicated, with every component needing to be custom-designed, the work is technically sound and timely. Considering an economically viable LIDAR system is expected to have a price tag in the \$100-1000 range, and this multi-chip laser system using III-V, SiN, SiGe, will be very expensive, there will be a long way to go for this source to be used in practical system. Nevertheless, I support publication in Nature Communications.

Minor comments:

1. On page 12, line 243, "Fig. 4c, d" should be "Fig. 3c, d."
2. On page 7, line 144, it alludes that the requirement of "waveform predistortion and linearization" could be a disadvantage. But those signal process techniques are routines and can be easily done in electronics. This point needs more discussion.

Reviewer #2 (Remarks to the Author):

The authors have demonstrated a fully-integrated FMCW laser source for coherent Lidar. Integrated external-cavity laser source, electrical driving chip, and integrated optical amplifier are combined to provide a fully integrated source, which is urgently demanded in present Lidar system. However, individual components have been reported previously, which weakens the novelty. Hybrid integration of the external cavity laser and the semiconductor optical amplifier for high output powers (220 mW) has been demonstrated in [1]. While considering this work demonstrates a fully-integrated FMCW source, which is essential for future chip-scale coherent Lidar system, this work can be published in Nature Communications after major revision. The technology scheme proposed and demonstrated in this work need more comprehensive discussion and comparison, to show the novelty.

[1] Chen C, Wei F, Han X, et al. Hybrid integrated Si₃N₄ external cavity laser with high power and narrow linewidth[J]. Optics Express, 2023, 31(16): 26078-26091.

Below are some comments:

1. The external cavity laser employed in this work consisted of a RSOA and a SiN external cavity, tuning through stress-optic effect. Integrated external cavity laser is a well-developed topic. It is suggested to give a detailed comparison of different external cavity laser schematics to show the novelty or validity of this choice.

2. The claim of better performance of EDWA over SOA in page 14 is quite confusing for me.

2.1 As the author arranged the EDWA at the emission part, for FMCW lidar, the modulation is carried out in frequency domain while the output power may stay steady ideally. I am wondering whether the gain nonlinearity of SOA is obviously worse than that of the EDWA, that could seriously degenerate the detected Lidar signal.

2.2 It is also hard for me to agree with that the worse temperature stability of SOA may cause obvious worse linearity compared with EDWA. As the ref. 34 only discusses about SOA and EDWA for lasers, it may not directly support this claim.

It is suggested for the authors to give some direct data, in simulation or in experiment to demonstrate the advantage of EDWA over SOA, as SOA is a common and commercially available optical amplifier.

3. The authors claim that the proposed laser is linearization-free. In line 214 page 10, the authors said: "we iteratively linearized the optical waveform employing a delayed homodyne detection method". Thus, an initial linearization is necessary. The linearization-free here refers that it does not require any further feedback while laser operation. Would the linearity degrade as the laser operation for a long period, for example a few days? Does the linearization need to be performed every turn-on process?

4. In this work, the authors adopt sawtooth wave for FMCW. It seems that this method could not measure the velocity based on Doppler frequency shift detection, which is one of the crucial advantages of coherent Lidar over ToF scheme.

5. The authors claim that their sources can "serve as a 'drop-in' solution in any FMCW LiDAR". It is suggested to show the ability of wideband wavelength tuning while carrying FMCW signals. This is fundamental for Lidar using dispersion elements for spatio-spectral mapping.

6. It is suggested to cite <https://arxiv.org/pdf/2308.15404.pdf> at Page 4, line 108, which may give a clear illustration of lower laser coherence by using electro-optic materials.

Reviewer #3 (Remarks to the Author):

This work aims to implement a photonic-electronic integrated circuit for a chip-scale LiDAR source. 3D ranging experiments with chirps of 2 GHz excursion, 50 kHz sweep rate, and 20 mW optical power are performed for a scene mapping at a distance of 10 meters. This result shows one of the latest successful implementations of an integrated coherent ranging source to replace existing bulk components, which has been a big obstacle to a coherent LiDAR engine. The paper is in general well written and the experimental data figures are also well organized. Overall, I recommend a publication in Nature Communications but would ask the authors to answer the following questions and comments:

Since the concept of the photonic-electronic LiDAR source in Fig. 1 is one of the main important parts of this work, I suggest the authors add more details to explain the connections among a few substrate chips. What are the connection structure and coupling efficiency between the RSOA III-V chip and the PZT Si₃N₄ chip ? What are the connection scheme and pumping efficiency between the EDWA Si₃N₄ chip and the separately required 1480 nm pump source ? How can be the input and output ports of each Si₃N₄ chip connected to the ends of external optical fibers ? What is the total volume size including most of the required components, such as HV-AWG ASIC chip, other optical chips, 1480 nm pumping source, and PCB board to place all of them on?

In Fig. 4 a, there are two 50:50 couplers to implement a Mach-Zehnder interferometer. Considering the transmission loss between reference and target paths, it should be further optimized to select the splitting ratio instead of 50 % and 50%.

The collimator lens used in this work can only ensure collimation within a short range. Can it still ensure that no obvious caustics occur at further distance ? Considering the imaging of the coherent LiDAR in your method, it is important to address how data collection and any jitter during the process are managed.

What is the maximum available imaging repetition rate for a given pixels based on the sweep rate of 50 kHz because the relatively low acquisition rate has long been the main bottleneck for coherent LiDARs compared with ToF LiDARs ?

There are minor errors to correct:

In the caption of Fig 3 e, the experimental setup is located in the right bottom inset, not in the left bottom inset as described with error.

In the sentence 'The optical spectra before and after the amplification stage are depicted in Fig. 4c,d, respectively.' The expression of 'Fig. 4c,d' should be replaced by 'Fig. 3c,d'

There are many acronyms without showing the initial letters, such as PZT, BTO, CMOS.

**Resubmission of a revised version of
Nature Communications Manuscript Number NCOMMS-23-39189**

“Photonic-electronic integrated circuit-based coherent LiDAR engine”

We are grateful that reviewers have seen our manuscript. After reading the comments made by the referees thoroughly, we would like to thank them for the detailed review and suggestions to improve the manuscript. We appreciate that the reviewers gave a positive evaluation such as:

“These performances are on par with the state-of-the-art FMCW demonstration realized with discrete optical and electronic components and systems. The demonstrated results are compelling and self-evident and the analysis is carefully done with standard approaches...” (Reviewer #1)

“This result shows one of the latest successful implementations of an integrated coherent ranging source to replace existing bulk components, which has been a big obstacle to a coherent LiDAR engine. The paper is in general well written and the experimental data figures are also well organized.” (Reviewer #3)

We have addressed the concerns and suggestions raised by the Referees, and formulated our point-by-point response below.

To keep this report clear, the Referees' comments are colored black while our responses use **blue** and changes to the manuscript are colored **red**. Changes in the manuscript file are marked **blue**.

Reviewer #1 (Remarks to the Author):

Lukashchuk et al. report an integrated system of frequency-chirped laser for FMCW. Narrow linewidth lasers with fast and large excursion frequency chirping are necessary for coherent FMCW LIDAR but remain to be a challenge. The authors attack this challenge with an impressive engineering endeavor that integrates multiple components, including a semiconductor optical amplifier (SOA), SiN Vernier cavities with PZT piezoelectric MEMS tuning, an erbium-doped waveguide amplifier (EDWA), and a SiGe BiCMOS. The achieved performance is equally impressive, with a frequency excursion of 2 GHz in 23 microseconds, leading to a ranging resolution of 15 cm. These performances are on par with the state-of-the-art FMCW demonstration realized with discrete optical and electronic components and systems. The demonstrated results are compelling and self-evident and the analysis is carefully done with standard approaches, so there are really no technical comments I can provide. Although, in my opinion, the claim that the system is "plug-and-play" is an exaggeration, as the whole system is very complicated, with every component needing to be custom-designed, the work is technically sound and timely. Considering an economically viable LIDAR system is expected to have a price tag in the \$100-1000 range, and this multi-chip laser system using III-V, SiN, SiGe, will be very expensive, there will be a long way to go for this source to be used in practical system. Nevertheless, I support publication in Nature Communications.

Reply: We thank the referee for the careful study of our manuscript, constructive comments, and his/her positive recommendation.

Actions taken: We removed the "plug-and-play" phrase from the manuscript.

Minor comments:

1. On page 12, line 243, "Fig. 4c, d" should be "Fig. 3c, d."

We thank the referee for the very careful reading of our manuscript and for finding this omission.

Actions taken: We updated the reference to Figure 3 in the main text.

2. On page 7, line 144, it alludes that the requirement of "waveform predistortion and linearization" could be a disadvantage. But those signal process techniques are routines and can be easily done in electronics. This point needs more discussion.

Reply: We thank the referee for noting the necessity for the extra discussion. Indeed, linearization techniques are common for FMCW LIDARs based on DFBs or MEMS-VCSELs, however, typically they require additional optical interferometers, which in our case could be implemented on the same Si₃N₄ chip. We added a discussion of linearization methods in the main text.

Actions taken: We added extra discussion on linearization techniques for FMCW LiDAR: "Optoelectronic feedback loop is usually utilized to lock the optical chirp detected via delay interferometer to an electronic reference^{1,2}. Other techniques involve preliminary iterative predistortion and linearization of the waveform^{3,4} or resampling of the recorded signal, all of which rely on auxiliary interferometers^{5,6}."

Reviewer #2 (Remarks to the Author):

The authors have demonstrated a fully-integrated FMCW laser source for coherent Lidar. Integrated external-cavity laser source, electrical driving chip, and integrated optical amplifier are combined to provide a fully integrated source, which is urgently demanded in present Lidar system. However, individual components have been reported previously, which weakens the novelty. Hybrid integration of the external cavity laser and the semiconductor optical amplifier for high output powers (220 mW) has been demonstrated in [1]. While considering this work demonstrates a fully-integrated FMCW source, which is essential for future chip-scale coherent Lidar system, this work can be published in Nature Communications after major revision. The technology scheme proposed and demonstrated in this work need more comprehensive discussion and comparison, to show the novelty.

[1] Chen C, Wei F, Han X, et al. Hybrid integrated Si₃N₄ external cavity laser with high power and narrow linewidth[J]. Optics Express, 2023, 31(16): 26078-26091.

Reply: We thank the referee for the careful study of our manuscript, the helpful corrections and constructive comments, and his/her positive recommendation.

We would like to comment on “However, individual components have been reported previously, which weakens the novelty.” Indeed, while separately EDWA and Vernier ECL have been reported previously, we point out that the demonstrated ASIC has a novel architecture, which has not been demonstrated before. It consists of a voltage-controlled ring oscillator which drives charge-pump circuits, where controlling the oscillation frequency in time allows generating of high-voltage arbitrary waveforms. Although charge pump circuit behavior is well-known, the ASIC in this work shows for the first time their use as an HV-AWG. Comparison to alternative HV-AWG methods is shown in SI Fig.1, where the proposed main advantage is generating arbitrary high-voltage waveforms at only the CMOS supply voltage. Since the ASIC in this work is implemented entirely in an advanced CMOS node, an additional level of miniaturization of the FMCW LiDAR source is achieved when the laser is driven by the HV-AWG ASIC. Using compact erbium doped waveguide amplifier on a Si₃N₄ chip

We note that Chen C. et al. (2023) appeared after the initial submission of our manuscript and during the review process. We cite⁷ in the updated version of our manuscript.

Actions taken: We added extra discussion for the ASIC to emphasize the novelty. We added the citation Chen C. et al. (2023)⁷.

Below are some comments:

1.The external cavity laser employed in this work was consisted of a RSOA and a SiN external cavity, tuning through stress-optic effect. Integrated external cavity laser is a well-developed topic. It is suggested to give a detailed comparison of different external cavity laser schematics to show the novelty or validity of this choose.

Reply: We note that integrated lasers are not the main focus of this article. We demonstrate FMCW LiDAR technology using fast tunable ECL and in an introduction to our laser source, we talk about different lasers employed for LiDAR including different photonic integrated approaches. We agree with the reviewer on the importance of such comparison and we added a discussion citing other approaches. We considered the following ECL schematics for fast-tunable integrated lasers: 1) DFB laser self-injection locked (SIL) to high-Q Si₃N₄ microresonators with an integrated piezoelectrical actuator. In the SIL approach, the frequency tuning range is typically limited by the injection locking range which is determined by the backreflection and Q-factor of the microresonator, and in our experiments the locking range is 1-2 GHz for Si₃N₄ microresonators. Tight injection locking is needed for high tuning linearity in this case; 2) Extended DBR lasers with gratings implemented in photonic chip enhanced with integrated piezoelectrical actuator⁸. In our experiments, the tuning range was limited by 1 GHz. Linearity and tuning range were determined by the fabrication quality of the piezoelectrical layer (AlN); 3) ECL with Vernier ring filter (this work) presents a solution which is less sensitive to fabrication imperfections in comparison to gratings and provides a larger tuning range which is limited by the saturation of PZT/AlN. In our approach, we were able to obtain broadband tuning with microheaters and fast, linear tuning with piezoactuators.

Actions taken: We added extra discussion in the introduction to the laser source. We cite⁴ and ⁸, where the comparative performance of the recent integrated laser approaches is summarized.

2. The claim of better performance of EDWA over SOA in page 14 is quite confusing for me.

2.1 As the author arrange the EDWA at the emission part, for FMCW lidar, the modulation is carried out in frequency domain while the output power may stay steady ideally. I am wondering whether the gain nonlinearity of SOA is obviously worse than that of the EDWA, that could seriously degenerate the detected Lidar signal.

2.2 It is also hard for me to agree with that the worse temperature stability of SOA may cause obvious worse linearity compared with EDWA. As the ref. 34 only discussing about SOA and EDWA for lasers, it may not directly support this claim.

It is suggested for the authors to give some direct data, in simulation or in experiment to demonstrate the advantage of EDWA over SOA, as SOA is a common and commercially available optical amplifier.

Reply: We thank the reviewer for bringing up this concern, and try to improve the clarification.

In reply to Q2.1: In the discussion section of EDWAs' potential advantages, we aimed to briefly introduce the general advantageous aspects of erbium-based gain over SOAs, such as low gain nonlinearity or cross-talk, low ASE noise, and temperature stability, rather than claiming the benefits specifically for the FMCW LiDAR. These advantages are widely recognized (DOI: 10.1109/JLT.2003.822144; DOI: 10.1109/JSTQE.2014.2351811). In the submitted manuscript, we envisaged these advantages

can be potentially beneficial to our FMCW LiDAR, in which unwanted, accompanied amplitude modulations usually occur, due to the varied resonance detuning while modulating the frequency. Indeed, there is no strong evidence to consider SOA nonlinearity worse when the input power is constant, even though SOA was shown to introduce nonlinear phase noise to phase-modulated signals in telecommunications experiments(DOI: 10.1109/JLT.2009.2036868, DOI: 10.1109/JQE.2005.843943]).

In reply to Q2.2: Although we envisaged that the well-known temperature stability of EDWAs can be favourable for our LiDAR, we so far do not have experimental data to support this claim, which is not the focus of this study. Therefore, we revised our statement and retracted our assertion regarding temperature stability and SOA gain nonlinearity.

Actions taken: 1) We added clarification that we foresee EDWAs might show advantages in terms of less gain nonlinearity in our FMCW LiDAR source that exhibits parasitic amplitude fluctuation. 2) we removed the phrase regarding temperature stability and SOA gain nonlinearity.

3. The authors claim that the proposed laser is linearization-free. In line 214 page 10, the authors said: "we iteratively linearized the optical waveform employing a delayed homodyne detection method". Thus, an initial linearization is necessary. The linearization-free here refers that it does not require any further feedback while laser operation. Would the linearity degrade as the laser operation for a long period, for example a few days? Does the linearization need be performed every turn-on process?

Reply: We appreciate the reviewer for this valuable comment. Indeed, the initial linearization routine is essential for determining the electrical waveform itself. Subsequently, this waveform is stored in memory and reapplied as needed. We have observed no degradation in nonlinearity during continuous laser operation over the course of a day, including when the laser is switched off and on with the same operational parameters (such as RSOA current and the current applied to the microring microheaters used to align the Vernier filter resonances). However, we have observed that making slight adjustments to the laser's operating point can result in a minor degradation of linearity. This effect can be mitigated, and the linearity improved to less than 0.1%, by repeating the calibration procedure.

Below in figure 1, we attach the linearity analysis of the laser performance on the two consecutive days when the laser was operated at the same parameters (after being switched OFF/ON). Both distributions feature exponential tails with mean nonlinearity below 0.1%.

Actions taken: We added extra discussion in Methods on linearity stability.

Figure 1. Tuning nonlinearity of Vernier ECL measured at different time.

4. In this work, the authors adopt sawtooth wave for FMCW. It seems that this method could not measure the velocity based on doppler frequency shift detection, which is one of the crucial advantages of coherent Lidar over ToF scheme.

Reply: Indeed, for the current waveform choice the beatnote frequency shift both due to the distance and velocity is entangled, i.e. for one value of the measured frequency shift there are two unknowns: distance and velocity. However, since it is a coherent measurement, the actual Doppler shift will be still measured. Even though we did not perform velocity measurement in our manuscript, the current setup with AWG and laser is not limited to this case. Adding a different chirp slope for every 2nd (even) ramp would give two frequency shifts for two unknowns, thus allowing us to infer the velocity.

The chirp of the opposite slope would result in a conventional triangular frequency modulation. Our AWG is capable of triangular waveform generation. However, linearization of the triangular waveform is challenging which is the limiting factor of the programmability of the AWG in its current architecture.

5. The authors claims that their sources can "serve as a 'drop-in' solution in any FMCW LiDAR". It is suggested to show the ability of wideband wavelength tuning while carrying FMCW signals. This is fundamental for Lidar using dispersion elements for spatio-spectral mapping.

Reply: Indeed, while focal plane architectures are free from the requirement of wideband tuning, optical phased array or other dispersive elements implementations require that. Using integrated heaters we demonstrate tuning within a 9 nm span around 1550 nm in Figure 2 of this reply. We foresee an increase in tuning by having a dedicated heater in SiN buswaveguide, while currently employed heaters are integrated on top of microresonators.

Actions taken: We added the wide tuning capability to the supplementary information.

Figure 2. Optical spectra of Vernier laser emission at different microheater power with 30 mW electrical power steps

6. It is suggested to cite <https://arxiv.org/pdf/2308.15404.pdf> at Page 4, line 108, which may give a clear illustration of lower laser coherence by using electro-optic materials.

Reply: We thank the reviewer for this note.

Actions taken: We added “We note Zhang et al. \cite{Zhang2023} have recently demonstrated that such materials exhibit significantly lower levels of cavity noise at high offset frequencies and are predominantly limited by thermal-charge-carrier-refractive noise”

Reviewer #3 (Remarks to the Author):

This work aims to implement a photonic-electronic integrated circuit for a chip-scale LiDAR source. 3D ranging experiments with chirps of 2 GHz excursion, 50 kHz sweep rate, and 20 mW optical power are performed for a scene mapping at a distance of 10 meters. This result shows one of the latest successful implementations of an integrated coherent ranging source to replace existing bulk components, which has been a big obstacle to a coherent LiDAR engine. The paper is in general well written and the experimental data figures are also well organized. Overall, I recommend a publication in Nature Communications but would ask the authors to answer the following questions and comments:

Reply: We thank the referee for the careful study of our manuscript, the helpful corrections and constructive comments, and his/her positive recommendation.

Since the concept of the photonic-electronic LiDAR source in Fig. 1 is one of the main important parts of this work, I suggest the authors add more details to explain the connections among a few substrate chips. What are the connection structure and coupling efficiency between the RSOA III-V chip and the PZT Si₃N₄ chip ?

Reply: In our experiment, RSOA is edge coupled to the Si₃N₄ chip where the coupling waveguide angle of the Si₃N₄ chip is optimized (in simulations) to achieve 20% power transmission (laser output power / nominal RSOA power).

Actions taken: We have updated the Methods section to describe the RSOA-Si₃N₄ chip coupling section.

What are the connection scheme and pumping efficiency between the EDWA Si₃N₄ chip and the separately required 1480 nm pump source ?

Reply: We use lensed fibres to couple the light into the EDWA chip. The 1480 nm pump laser diode is connected via the 1480/1550 nm coupler. We are currently working on implementing a butt-coupled 1480 nm pump diode with the 1480/1550 nm splitter realized on the same Si₃N₄ chip as EDWA. We have demonstrated (Liu et al. Science 2022) that our EDWA can reach 145 mW on-chip signal power for an input power of 2.61 mW at a 245 mW pump, in which we achieved an on-chip power conversion efficiency of 60%. However, when including the excess insertion losses from the chip-fiber optical coupling and other fiber components, we estimated a wall-plug efficiency of < 10%.

Actions taken: We have updated the Methods section.

How can be the input and output ports of each Si₃N₄ chip connected to the ends of external optical fibers ?

Reply: We use a fully packaged solution for the laser chip. As depicted in Fig 1C of the main manuscript, the RSOA chip is butt-coupled and glued, and high numerical aperture fiber is glued to the output waveguide of the laser chip.

As for the EDWA chip, we use lensed fibers to couple the light in/out. We also highlight in the discussion that the design of EDWA on the same chip as an external cavity laser waveguide is possible and would greatly simplify the setup.

What is the total volume size including most of the required components, such as HV-AWG ASIC chip, other optical chips, 1480 nm pumping source, and PCB board to place all of them on?

Reply: The ASIC is 1.2 mm x 1.1 mm, and its thickness is a few hundred micrometers. The driving PCB is 9 cm x 7.5 cm, with a height of around 1.5 cm which includes the ASIC chip. The laser chip is enclosed in a standard butterfly 14 PIN package with dimensions: 15 mm x 30 mm x 10 mm. We used a butterfly-packaged 1480 nm pump laser with a commercial compact laser mount of similar size. However, we envision the co-integration of both Si₃N₄ chips on the same PCB (or having Si₃N₄ laser waveguide and Erbium-doped waveguide on the same chip as discussed earlier) with the ASIC chip. In that case, the total volume would be less than 10cm x 10cm x 5cm.

In Fig. 4 a, there are two 50:50 couplers to implement a Mach-Zehnder interferometer. Considering the transmission loss between reference and target paths, it should be further optimized to select the splitting ratio instead of 50 % and 50%.

Reply: The second 50/50 splitter is due to the balanced detection, which allows for the common mode rejection and optimal heterodyne detection. The motivation for the first 50/50 splitter is governed by the need to have a strong enough LO power to have shot noise-limited detection. For the laser output equal to 3mW we had ~1mW of optical power in the LO path.

The collimator lens used in this work can only ensure collimation within a short range. Can it still ensure that no obvious caustics occur at further distance ? Considering the imaging of the coherent LiDAR in your method, it is important to address how data collection and any jitter during the process are managed.

Reply: We thank the reviewer for noting this important issue. Indeed, the impact of atmospheric turbulence can not be neglected in constructing real LiDAR and it is a huge research topic in itself. We note it in the manuscript and cite the relevant literature. We note that surprisingly for a monostatic configuration, enhancement in heterodyne collection efficiency and SNR is possible compared with turbulent-free light propagation^{9,10}.

We used a standard Thorlabs fiber collimator with an aperture size ~1cm collimated to infinity. The ranging distance of 10m was limited by the lab dimensions (see the photo in SI). Commercial coherent LiDARs employ collimator apertures of ~3cm to ensure low angular divergence and high irradiance of the target for ≥100m ranging.

Actions taken: We added “We note that in practical LiDAR applications, it is essential to consider real-world factors such as atmospheric turbulence, including variations in the refractive index along the beam path, as well as the influence of transverse wind ^{9,10}.”

What is the maximum available imaging repetition rate for a given pixels based on the sweep rate of 50 kHz because the relatively low acquisition rate has long been the main bottleneck for coherent LiDARs compared with ToF LiDARs ?

Reply: The imaging repetition rate will be governed by the number of pixels in a frame and total acquisition speed. In our case, we acquired one frame of 10800 pixels in a triangular scan by galvo-mirrors during 0.25 seconds. That said, for frames of 100x100 pixels with 50 kHz acquisition rate it will take 0.25s to acquire the whole frame. Indeed, as the reviewer noticed this is a problem in a coherent LiDAR community. ToF LiDAR solves it by employing parallelization of both lasers and detectors up to 256 units (Ouster, Velodyne). That is possible by utilizing cheap legacy components. Currently, the high cost of coherent frequency-agile lasers impedes the same approach. We strongly believe integrated photonics will tackle this problem and bring maturity to FMCW LiDAR.

There are minor errors to correct:

In the caption of Fig 3 e, the experimental setup is located in the right bottom inset, not in the left bottom inset as described with error.

In the sentence ‘The optical spectra before and after the amplification stage are depicted in Fig. 4c,d, respectively.’ The expression of ‘Fig. 4c,d’ should be replaced by ‘Fig. 3c,d’

There are many acronyms without showing the initial letters, such as PZT, BTO, CMOS.

We would like to express our gratitude to the reviewer for their meticulous review.

Actions taken: We have addressed the identified errors and incorporated descriptions for the acronyms in the main manuscript.

1. Satyan, N., Vasilyev, A., Rakuljic, G., Leyva, V. & Yariv, A. Precise control of broadband frequency chirps using optoelectronic feedback. *Opt. Express, OE* **17**, 15991–15999 (2009).
2. Behroozpour, B. *et al.* Electronic-Photonic Integrated Circuit for 3D Microimaging. *IEEE Journal of Solid-State Circuits* **52**, 161–172 (2017).

3. Zhang, X., Pouls, J. & Wu, M. C. Laser frequency sweep linearization by iterative learning pre-distortion for FMCW LiDAR. *Opt. Express, OE* **27**, 9965–9974 (2019).
4. Lihachev, G. *et al.* Low-noise frequency-agile photonic integrated lasers for coherent ranging. *Nat Commun* **13**, 3522 (2022).
5. Xi, J., Huo, L., Li, J. & Li, X. Generic real-time uniform K-space sampling method for high-speed swept-Source optical coherence tomography. *Opt. Express, OE* **18**, 9511–9517 (2010).
6. Okano, M. & Chong, C. Swept Source Lidar: simultaneous FMCW ranging and nonmechanical beam steering with a wideband swept source. *Opt. Express, OE* **28**, 23898–23915 (2020).
7. Chen, C. *et al.* Hybrid integrated Si₃N₄ external cavity laser with high power and narrow linewidth. *Opt. Express, OE* **31**, 26078–26091 (2023).
8. Siddharth, A. *et al.* Hertz-linewidth and frequency-agile photonic integrated extended-DBR lasers. Preprint at <https://doi.org/10.48550/arXiv.2306.03184> (2023).
9. Feneyrou, P. *et al.* Frequency-modulated multifunction lidar for anemometry, range finding, and velocimetry—1 Theory and signal processing. *Applied Optics* **56**, 9663 (2017).
10. Frehlich, R. G. & Kavaya, M. J. Coherent laser radar performance for general atmospheric refractive turbulence. *Appl. Opt., AO* **30**, 5325–5352 (1991).

REVIEWERS' COMMENTS

Reviewer #1 (Remarks to the Author):

I'm satisfied with the authors' response and revision, thus support publication in the current stage.

Reviewer #2 (Remarks to the Author):

The authors have addressed all my concerns. I believe it is ready for publication.

Reviewer #3 (Remarks to the Author):

The revised manuscript is improved with the reasonable improvements. This report shows one of the successful implementations of an integrated coherent ranging source to replace existing bulk components of a coherent LiDAR engine.

Resubmission of a revised version of
Nature Communications Manuscript Number NCOMMS-23-39189
“Photonic-electronic integrated circuit-based coherent LiDAR engine”

We are grateful that reviewers have seen our revised manuscript. After reading the comments made by the referees thoroughly, we would like to thank them for their positive assessment and recommendation of our work.

Dr. habil. Tobias J. Kippenberg (PhD)
Full Professor
EPFL – Swiss Federal Institute of
Technology Lausanne
Personal assistant: Kathleen Vionnet
(+ 41 21 693 44 52)
Grant management: Antonella Ragnelli
(+ 41 21 693 44 17)